# Adolescent Athlete Engagement and Team Cohesion in Football: A Moderated Mediation Model with Gender-Based Insights

**DOI:** 10.3390/bs15091264

**Published:** 2025-09-16

**Authors:** Bingzhi Wan, Huarui Huang, Xiaoqi Sha, Chen Zhong, Yizhou Shui

**Affiliations:** 1Department of Physical Education, Xidian University, Xi’an 710126, China; 18691306913@snnu.edu.cn; 2Teacher Development College, Shaanxi Normal University, Xi’an 710062, China; 3School of Physical Education, Shaanxi Normal University, Xi’an 710119, China; huanghuarui1102@163.com (H.H.); shaxiaoqi@snnu.edu.cn (X.S.); zhongchen1207@163.com (C.Z.); 4School of Psychology, Shaanxi Normal University, Xi’an 710062, China

**Keywords:** football, adolescents, group dynamics, moderated mediation, gender differences

## Abstract

Adolescents often face interpersonal and adjustment challenges when transitioning from a family-centered to a school-based environment, especially without a supportive group climate. To address these challenges, this study used football, the world’s most widely played team sport, as a platform to examine the impact of athlete engagement on team cohesion and its underlying mechanisms. A total of 1692 Chinese adolescents who regularly participated in football training and demonstrated a strong passion for the sport were recruited. Data were collected using the Athlete Engagement Questionnaire (AEQ), the Interpersonal Competence Questionnaire (ICQ), the Perceived Workplace Social Support Scale (PWSSS), and the Group Environment Questionnaire (GEQ), all of which demonstrated good reliability and validity in this study. The results revealed that (1) athlete engagement was positively associated with team cohesion team cohesion; (2) interpersonal competence partially mediated the relationship between athlete engagement and team cohesion; (3) social support moderated both the direct relationship between athlete engagement and team cohesion and the indirect relationship between athlete engagement and interpersonal competence; and (4) social support moderated the relationship between athlete engagement and team cohesion with significant gender differences, whereas no gender differences were observed in the relationship between athlete engagement and interpersonal competence. This moderated mediation model not only enriches the conceptual model of group cohesion but also addresses gaps in the current literature. Furthermore, it provides theoretical support for physical educators to design targeted team sports interventions tailored to the characteristics of different gender groups.

## 1. Introduction

Cohesion has been defined as a dynamic process reflected in a group’s tendency to remain united in pursuit of instrumental goals and the satisfaction of members’ emotional needs ([71]). Team cohesion is closely linked with positive youth development ([7]). Excellent team cohesion is essential for team success ([5]), and it also plays an important role in success across other fields ([57]). Adolescents undergo marked environmental transitions in both sport and daily life. Early childhood activities are largely family-centered ([50]), whereas during adolescence the school becomes the primary context for living and sport participation ([32]). When a positive group climate is absent, students are more likely to experience exclusion and neglect. Conflicts within the team and loneliness may also occur. It has been demonstrated that well-functioning sports teams not only provide superior team results ([11]), but also foster greater personal growth of adolescents ([7]). Current research indicates that team cohesion and sport performance have a mutual relationship ([3]; [39]), particularly in football ([77]). However, a lack of discussion on the mechanisms and conditions underlying this relationship has been observed, as objective performance is influenced by numerous factors ([27]). To address this gap, some researchers have proposed that athlete engagement can serve as an objective measure of sports performance and predict team cohesion ([27]; [77]).

Compared with individual sports, team sports often yield better social and mental outcomes ([20]; [41]). This advantage is likely attributable to the social nature and positive experiences of team sports, such as peer support and a sense of belonging ([21]). Through teamwork, team sports enable adolescents to learn effective communication and manage interpersonal relationships. They also help build friendships and strengthen social support networks, which gradually foster self-worth and confidence in the team environment ([20]; [41]). Over time, the ongoing cooperation and interaction among teammates cultivate a sense of closeness and cohesion among children and adolescents who participate in team sports ([22]). Given the global scale of football participation and its popularity among adolescents, athlete engagement in football is often used as an indicator of attitudes toward sport and its effects, and as a predictor of adolescents’ social and mental functioning ([30]). Although prior research has shown that athlete engagement positively predicts team cohesion in football (e.g., [77]), the contribution of the distinctive social attributes of team sports to this association has not been fully explained. Specifically, [77] ([77]) developed a mediation model in a football context and found that football participation did not directly enhance team cohesion; instead, athlete engagement was the factor driving cohesion. However, they did not further examine how social attributes unique to team sports, such as interpersonal relationships and peer support, contribute to this process. It is still unclear whether social factors in team sports mediate or moderate the association between athlete engagement and team cohesion.

According to the conceptual model of group cohesion, team cohesion comprises two dimensions, namely task cohesion and social cohesion ([68]). Task cohesion relates to the willingness of team members to work toward shared goals, whereas social cohesion depends on interpersonal relationships and social support among team members ([68]). Meanwhile, gender is also an important factor influencing team cohesion ([11]). Therefore, this study adopts the conceptual model of group cohesion as the theoretical framework to analyze the mechanisms through which athlete engagement relates to team cohesion, with a focus on social factors in team sports, thereby addressing prior research gaps. Specifically, the study examines whether interpersonal competence and social support mediate or moderate the association between athlete engagement and team cohesion, and whether these relationships differ by gender. We aim to extend the conceptual model of group cohesion to the adolescent football context and to generate practice-oriented evidence on how social factors strengthen cohesion in adolescent teams. At a practical level, we seek to use team sport to cultivate cohesion among adolescents and to support their growth within a positive team climate.

### 1.1. Athlete Engagement and Team Cohesion

Athlete engagement is defined as an enduring positive cognitive and emotional experience in sports, with confidence, dedication, vigor, and enthusiasm as the main characteristics ([44]). Prior studies have shown that athlete engagement in football can influence team cohesion ([77]). One of the most common reasons for participating in football is the desire to be part of a team. To be called a “good team player” seems to connote something nobler than “a talented player” or even “a well-trained player” ([25]). Additionally, according to the conceptual model of group cohesion, task cohesion relates to the desire of team members to work toward shared goals ([68]). This requires individual athletes on a team to be confident not only in their own abilities but also in their teammates’ ability to contribute to the team’s goals ([51]). These are integral components of athlete engagement. Therefore, Hypothesis 1 (**H1**) is proposed.

**H1.** 
*Athlete engagement is positively associated with team cohesion.*


### 1.2. The Mediation of Interpersonal Competence

Interpersonal competences are those which we use in relations with other people and they are tied closely associated with effective functioning in interpersonal relationships ([38]). Interpersonal competences are acquired, shaped, and refined through learning ([38]). Compared with other activities, sports activities pay more attention to the rules of games and teamwork to cultivate students’ communication skills and interpersonal relationship-handling methods ([72]). Team sports, in particular, provide adolescents with increased opportunities for peer interaction ([41]). At the same time, interpersonal relationship plays a crucial role in team cohesion ([62]). According to the conceptual model of group cohesion, social cohesion depends on interpersonal relationships among team members ([68]). That is, interpersonal competence serves as a proximal process that translates individual athlete engagement into stronger cohesion. The previous study shows that the cohesion of a sports team is associated with the strength of the relationships between coaches and participants as well as between participants ([64]). Football is characterized by frequent interaction during training and competition, team identity, and recognition of individual contributions, all of which are associated with stronger team cohesion ([77]). When athlete engagement elevates interpersonal competence in team settings, the group is more likely to organize around shared goals and to consolidate supportive ties, which is consistent with the conceptual model of group cohesion’s pathways to cohesion. Therefore, **H2** is proposed.

**H2.** 
*Interpersonal competence plays a mediating role between athlete engagement and team cohesion.*


### 1.3. The Moderation of Social Support

Social support refers to an individual’s perception of the availability of help or support from others in their social network ([4]). Social support has an important association on the development of football participants among adolescents and is considered as an indispensable factor ([53]). Although the conceptual model of group cohesion indicates that social support influences team cohesion ([68]), the precise mechanisms underlying this relationship remain unclear. The previous study shows that social support plays a moderating role in the relation between physical exercise and social anxiety ([54]). Individuals with higher perceived social support have more sufficient mental resources to handle interpersonal relationship ([54]). In other words, strengthening social support appears to reinforce the key factors through which athlete engagement affects social cohesion, namely communication and interpersonal relationships. On the other hand, on more cohesive teams, members expect to receive more care, encouragement, instruction, and assistance ([70]). Peer social support has been identified as a key factor influencing team cohesion ([65]). In addition, confidence in peer leadership and the support it generates has been shown to strengthen team cohesion ([51]). Coaches who provide higher levels of social support also promote stronger task cohesion among athletes ([24]). Therefore, **H3a** and **H3b** are proposed.

**H3a.** 
*Social support moderates the direct prediction effect athlete engagement on team cohesion.*


**H3b.** 
*Social support moderates the mediating effect of interpersonal competence (Figure 1).*


### 1.4. Gender Differences

According to social role theory, society assigns different expectations and roles to individuals based on their gender, which influences their behavior and attitudes ([16]; [17]). Prior research has shown gender difference in adolescent football, with male adolescents, compared with female adolescents, often displaying stronger competitive and recreational orientations6 ([73]). In addition, previous studies have shown that team cohesion is correlated with gender differences ([2]), but few previous studies have focused on gender differences in interpersonal and social support ([74]). Females and males have different concerns regarding interpersonal relationship: females tend to establish binary relationships and value individual intimacy and self-disclosure, while males prefer to establish friendships by participating in group activities and developing large group relationships ([15]). Additionally, females tend to perform more positive qualities in teacher–student relationships. In terms of social support, there were notable distinctions between the genders as well. For example, females are more likely to approach others to ask for social assistance. Therefore, **H4a** and **H4b** are proposed.

**H4a.** 
*There are gender differences in the moderating effect of social support on the association between athlete engagement and team cohesion.*


**H4b.** 
*There are gender differences in the moderating effect of social support on the association between athlete engagement and interpersonal competence.*


## 2. Methods

### 2.1. Participants

This cross-sectional study recruited 1692 adolescents from 64 schools in 5 provinces of China using a convenience sampling approach. A total of 33 adolescents discontinued participation for personal reasons, yielding 1659 valid questionnaires. Schools that offered organized extracurricular football were contacted through their administrations, and participation was confirmed by the school head or a physical education teacher. Eligible participants were adolescents aged 8–16 years. Physical education staff identified students who met the following inclusion criteria: at least one year of organized football training and participation in football as a leisure activity rather than as professionals. Students not engaged in football training were excluded. Entire squads were invited to participate to minimize selection at the coach or student level, and all eligible students present on the survey day were approached. Information sheets were distributed in advance to students and guardians, and only those who returned signed guardian consent and student assent were enrolled.

### 2.2. Procedure

Ethical clearance was granted by the Ethics Committee of Shaanxi Normal University (Approval No. 202416012). School administrations authorized the survey, and written guardian consent together with student assent were obtained prior to data collection. Participants were briefed on the study aims, procedures, and confidentiality measures, and all procedures adhered to the principles of voluntariness and confidentiality. Surveys were completed in classrooms under the supervision of trained researchers, and participants could withdraw at any time. Responses were recorded without personal identifiers, and all data were kept confidential and used solely for scientific research.

### 2.3. Measures

#### 2.3.1. Athlete Engagement Questionnaire (AEQ)

Athlete engagement was measured with the AEQ developed by Lonsdale and Jackson ([44]). We used Zhang’s Chinese adaptation of the AEQ ([75]), for which studies in Chinese adolescent samples have reported satisfactory reliability and validity ([26]). The instrument contains 16 items covering 4 facets—self-confidence, vigor, dedication, and enthusiasm—rated on a 5-point Likert scale (1 = never to 5 = always). One example of an item is as follows: “I am full of passion in training and competition.” The results of confirmatory factor analysis results show that χ^2^/df = 8.422, RMSEA = 0.067, AGFI = 0.913, GFI = 0.939, NFI = 0.955, IFI = 0.961, TLI = 0.950, and CFI = 0.960. In this study, Cronbach’s α of the AEQ was 0.948.

#### 2.3.2. Interpersonal Competence Questionnaire (ICQ)

This questionnaire was developed by Buhrmester ([8]), and translated into Chinese by Huang ([31]). The questionnaire has been proven to have good reliability and validity ([60]). The Chinese version of the ICQ consists of 15 items. All items in the scale were rated on a 5-point Likert scale (from 1 = strongly disagree to 5 = strongly agree); the higher the score is, the higher the interpersonal competence ability. One example of an item is as follows: “Introduce yourself to someone you might want to meet or develop a relationship with.” The results of confirmatory factor analysis results show that χ^2^/df = 5.381, RMSEA = 0.051, AGFI = 0.948, GFI = 0.963, NFI = 0.958, IFI = 0.966, TLI = 0.958, and CFI = 0.966. In this study, Cronbach’s α of the measured ICQ was 0.913.

#### 2.3.3. Perceived Workplace Social Support Scale (PWSSS)

The PWSSS, developed by Brouwers and translated into Chinese by Ju ([6]; [34]), consists of 2 dimensions, namely instrumental support and emotional social support, with a total of 22 items. To make the measure more relevant to students, adjustments were made to the original questionnaire. The term “colleagues” was changed to “classmates”, and “leaders” was replaced with “teachers”. All items in the scale were rated on a 5-point Likert scale (1 = strongly disagree, 5 = strongly agree). One example of an item is as follows: “My classmates would support me.” The results of confirmatory factor analysis results show that χ^2^/df = 9.278, RMSEA = 0.071, AGFI = 0.873, GFI = 0.906, NFI = 0.932, IFI = 0.939, TLI = 0.924, and CFI = 0.939. In this study, the Cronbach’s α for the PWSSS was 0.960.

#### 2.3.4. Group Environment Questionnaire (GEQ)

The GEQ was developed by Carron ([10]). The original version of the GEQ consists of 16 items across 4 dimensions, namely individual attractions to the group—task (ATG-T), individual attractions to the group—social (ATG-S), group integration—task (GI-T), and group integration—social (GI-S), using a 7-point Likert scale. In China, the GEQ was localized and revised, reducing the number of items to 15 and adopting a 5-point Likert scale ([45]; [76]). One example of an item is as follows: “I would be willing to actively participate in our team’s internal bonding activities.” The results of confirmatory factor analysis results show that χ^2^/df = 4.249, RMSEA = 0.049, AGFI = 0.954, GFI = 0.967, NFI = 0.965, IFI = 0.972, TLI = 0.966, and CFI = 0.972. In this study, the revised GEQ was used, with a Cronbach’s α of 0.858.

### 2.4. Data Analysis

Apart from confirmatory factor analysis, which was estimated in AMOS 28.0, all data analyses were conducted in SPSS 27.0. First, Harman’s one-factor analysis was employed to check for common method biases. Second, descriptive statistics and correlations among the variables were assessed. Third, the mediation effect of interpersonal competence and the moderating role of social support were tested using Model 4 and Model 8 of the PROCESS macro for SPSS ([29]). An effect was considered significant if the 95% confidence interval (CI) did not include zero, based on a bootstrap random sample (*n* = 5000). To examine how social support moderates the relationships between athlete engagement, interpersonal competence, and team cohesion, a simple slope test was performed ([1]). An interaction diagram based on psychological detachment was also used. Finally, multi-group modeling was conducted to examine gender differences in the moderated mediation model.

## 3. Results

### 3.1. Basic Characteristics

A total of 1692 adolescents from primary school to senior high school participated in the study. After screening, 1659 valid questionnaires were collected, yielding a validity rate of 98.0%. The detailed characteristics of participants are shown in Table 1.

### 3.2. Common Method Variance

Harman’s single factor test was used to test for common method bias ([52]). The results showed that nine factors had eigenvalues greater than 1. The first factor explained 33.494% of the variance, which was less than the critical value of 40% ([43]). Furthermore, the results of confirmatory factor analysis results show that χ^2^/df = 14.720, RMSEA = 0.091, AGFI = 0.373, GFI = 0.410, NFI = 0.559, IFI = 0.576, TLI = 0.562, and CFI = 0.576. The model fitting showed unsatisfactory results, showing that there was no significant common method bias in this study.

### 3.3. Descriptive Statistics and Correlations Analyses

Table 2 and Table 3 presents the descriptive statistics of the variables. The correlation analyses revealed significant positive correlations between athlete engagement, interpersonal competence, social support, and team cohesion (all *p* < 0.001). In addition, to evaluate multicollinearity in the regression models with team cohesion as the dependent variable, we examined tolerance and variance inflation factors for each predictor. Athlete engagement had a tolerance of 0.546 and a variance inflation factor of 1.831; interpersonal competence had a tolerance of 0.554 and a variance inflation factor of 1.807; social support had a tolerance of 0.585 and a variance inflation factor of 1.710. All diagnostics were comfortably below conventional thresholds, with all variance inflation factors well under 5, indicating that multicollinearity was not a concern.

### 3.4. Mediation Analysis

The mediation effect was tested before assessing the moderation effects (Table 4). Athlete engagement was positively associated with both team cohesion (*β* = 0.262, *t* = 9.569, *p* < 0.001, CI = [0.127, 0.193]) and with interpersonal competence (*β* = 0.612, *t* = 31.189, *p* < 0.001, CI = [0.593, 0.672]). Furthermore, interpersonal competence was positively associated with team cohesion (*β* = 0.265, *t* = 9.666, *p* < 0.001, CI = [0.125, 0.188]). Therefore, interpersonal competence plays a partial mediating role in the relationship between athlete engagement and team cohesion (indirect effect = 0.099, Boot SE = 0.012, 95% CI = [0.076, 0.122]). This model accounted for 25.65% of the variance in team cohesion. Thus, **H1** and **H2** were supported.

### 3.5. Moderated Mediation Model Analysis

In the second step, we employed Model 8 in the SPSS extension macro (Model 8 moderates the direct path and the first stage of the mediation model, which aligns with the hypothetical model in this study), and tested the moderated mediation model. As shown in Table 4, after including social support in the model, the interaction between athlete engagement and social support was significantly associated with both interpersonal competence (*β* = 0.166, *t* = 9.063, *p* < 0.001) and team cohesion (*β* = 0.155, *t* = 6.967, *p* < 0.001). These results suggest that social support moderates both the associations between athlete engagement and team cohesion and between athlete engagement and interpersonal competence. Thus, **H3a** and **H3b** were supported.

To understand how the moderator works, simple slope analysis was carried out, as shown in Figure 2 ([1]). The relation between athlete engagement and team cohesion was more positive under a high level of social support (*M* + 1*SD*, *simple slope* = 0.577, *t* = 20.611, *p* < 0.001) than under a low level of social support (*M −* 1*SD*, *simple slope* = 0.301, *t* = 11.747, *p* < 0.001, Figure 2a). Hence, the results indicated that increasing the level of social support can increase the association between athlete engagement and team cohesion.

The relation between athlete engagement and interpersonal competences was more positive under a high level of social support (*M + 1SD*, *simple slope* = 0.377, *t* = 10.109, *p* < 0.001) than under a low level of social support (*M − 1SD*, *simple slope* = 0.118, *t* = 3.738, *p* = 0.002, Figure 2b). Therefore, it is suggested that an increase in social support leads to an increase in the prediction of athlete engagement on team cohesion. Hence, the results indicated that increasing the level of social support can increase the association between athlete engagement and interpersonal competence.

### 3.6. Gender Differences Analysis

In the third step, the subject groups were then categorized by gender and analyzed for gender differences by the moderated mediation model. Table 5 shows that, in the male group, the interaction between athlete engagement and social support was significantly associated with interpersonal competence (*β* = 0.157, *t* = 7.153, *p* < 0.001), but it was not significantly associated with team cohesion (*β* = 0.020, *t* = 0.825, *p* = 0.409).

Table 6 shows that, in the female group, the interaction between athlete engagement and social support was significantly associated with both interpersonal competence (β = 0.196, t = 5.250, *p* < 0.001) and team cohesion (β = 0.020, t = 8.325, *p* < 0.001). These results suggest that social support moderated the associations between athlete engagement and team cohesion with gender differences, while no gender differences were evident in the association between athlete engagement and interpersonal competence. Thus, **H4a** was supported, but **H4b** was not supported.

To understand how the moderator works in different gender groups, simple slope analysis was carried out, as shown in Figure 3 ([1]). In the male group, the relation between athlete engagement and interpersonal competences was more positive under a high level of social support (*M* + 1*SD*, *simple slope* = 0.561, *t* = 15.679, *p* < 0.001) than under a low level of social support (*M* − 1*SD*, *simple slope* = 0.264, *t* = 8.275, *p* < 0.001, Figure 3a).

In the female group, athlete engagement level sport played a more essential role in predicting level of team cohesion in high level of social support (*M* + 1*SD*, *simple slope* = 0.591, *t* = 12.669, *p* < 0.001), rather than low level of social support (*M* − 1*SD*, *simple slope* = 0.325, *t* = 7.002, *p* < 0.001, Figure 3b). Meanwhile, the relation between athlete engagement and interpersonal competences was more positive under a high level of social support (*M* + 1*SD, simple slope* = 0.425, *t* = 6.910, *p* < 0.001) than under a low level of social support (*M* − 1*SD*, *simple slope* = −0.763, *t* = −1.364, *p* = 0.173, Figure 3c).

## 4. Discussion

This study further enriches the conceptual model of group cohesion by constructing a moderated mediation model. We found associations indicating that athlete engagement is positively related to team cohesion, and both the mediating role of interpersonal competence and the moderating role of social support were significant in this association. Additionally, gender differences were observed in this moderated mediation model. Within the conceptual model of group cohesion, our results clarify how the distinctive social features of team sports contribute to the engagement–cohesion link. This study provides theoretical support for physical educators to design team sports interventions based on the characteristics of different gender groups. We aim to enhance team cohesion within adolescent groups through team sports, fostering the improvement of social adaptability and mental health in a positive team environment. By strengthening interactions among adolescents, we can effectively enhance their interpersonal competence and help them learn to support each other and make progress together. This not only helps increase adolescents’ sense of collective identity but also promotes individual adaptation and development in social terms.

### 4.1. Team Cohesion Is Positively Associated with Athlete Engagement

The results indicated that athlete engagement was positively associated with team cohesion. This finding supports **H1** and is consistent with previous similar studies ([77]). Athlete engagement is a mindset of fulfillment during practice ([42]). When the success expectations of team members increase, this expectation tends to relate to higher team cohesion ([40]), which reflects the conceptual model of group cohesion by illustrating how athlete engagement contributes to goal-oriented collaboration. In terms of social cohesion, football teams often develop stronger interpersonal bonds through collaborative processes, such as joint problem-solving and coordinated strategy development aimed at achieving shared objectives ([64]). Unlike the individualistic culture of the West, Chinese culture places greater emphasis on collectivism ([46]). Within collectivistic cultures, group interests and the collective good take precedence over individual interests ([46]). Individuals in such cultures are more likely to internalize the purpose, tasks, and principles of group activities into their behavioral standards, subsequently adjusting and adapting to the norms established by these standards ([27]). This collectivist mindset helps enhance team cohesion, as Chinese adolescents in sports not only focus on individual achievements but also on contributing to the realization of the team’s collective goals. Therefore, athlete engagement is positively associated with team cohesion.

### 4.2. The Mediating Effect of Interpersonal Competence

The results indicated that interpersonal competence partially mediated the association between athlete engagement and team cohesion, which is consistent with **H2** and previous study ([64]). In football training and competition, the players’ active participation and dedication not only improve the competitive level of the team, but also promote mutual understanding and trust among the players ([77]). This experience of working together helps players better appreciate their connections with each other, which is associated with closer team relationships and stronger team cohesion ([77]).

As for the majority of adolescents, school life is the most significant aspect of their lives ([19]; [59]). The main relationships that adolescents have at school are with their teachers and peers ([13]; [69]). School sport, including football, may provide adolescents with opportunities to bond with other students, and interact with their peers and coaches ([32]). In Chinese schools, the other role of physical education teachers is team coaches. From the perspective of the teacher–student relationship, teachers can offer support and guidance to participants during football training and competition, which can ultimately contribute to the formation of strong bonds between teachers and students ([33]). From the perspective of peers, participation team sports can enhance adolescents’ social skills by improving understanding and cooperation through interactions with peers, thereby fostering quality interpersonal relationships ([41]). Therefore, many participants tend to develop good friendships and good interpersonal relationship in a team to pursue a shared goal ([14]; [23]). When a team works towards a common goal, this process is often linked to higher team cohesion ([23]). This goal-oriented collaboration is closely tied to the formation of task cohesion, reflecting the collective commitment to achieving common objectives. Furthermore, team cohesion is positively influenced by stronger interpersonal relationship among team members and a shared agreement ([64]). This pattern aligns with theoretical expectations that social cohesion is constructed through interpersonal interactions and value alignment ([68]). Therefore, interpersonal competence links athlete engagement to both task and social facets of the conceptual model of group cohesion by improving communication, coordination, conflict management, and role clarity.

### 4.3. The Moderating Effect of Social Support

The results showed that social support moderated the association between athlete engagement and team cohesion, suggesting that this association was stronger for adolescents with higher levels of social support. This finding supported **H3a** and was consistent with a previous study ([61]). Young athletes rely heavily on coaches and teammates as sources of motivation, perceived competence, and skill information ([66]). Studies have shown that higher perceived social support and a democratic leadership style from coaches, as well as peer support, are all linked to greater task cohesion ([36]; [65]; [67]). Participants in a team may feel responsible for helping and encouraging their peers to achieve higher levels of social support, which, in turn, is associated with stronger team cohesion ([9]). Furthermore, when group members recognize the social support they receive and understand their roles in achieving team goals, they are more likely to integrate into the group, leading to stronger team cohesion ([71]).

Additionally, social support moderated the relationship between athlete engagement and interpersonal competence, indicating that social support moderated the mediating effect of interpersonal competence. Specifically, the association between athlete engagement and interpersonal competence was stronger for individuals with high social support than for those with low social support. This finding supported **H3b** and was consistent with previous study ([58]). Increased levels of social support are associated with fewer feelings of loneliness among adolescents ([48]). Meanwhile, adolescents who receive higher support from their teachers and peers tend to have better interpersonal relationship ([28]; [47]). In team sports, adolescents who have higher levels of social support tend to have better social skills, including responsibility and openness, which can effectively facilitate their friendships ([56]). Furthermore, the social relationships and support provided through sports can also offer participants a sense of belonging and companionship ([18]). This is especially important in football, where the progress of one member can lead to the progress of the entire team, thereby supporting the development of leadership and social skills (i.e., the ability to build effective relationships with others) ([55]).

Overall, within the conceptual model of group cohesion, social support functions as a contextual condition rather than a simple input. Specifically, two pathways are implicated. First, higher social support strengthens the direct association between athlete engagement and task cohesion by clarifying goals and roles and aligning effort with team objectives. Second, higher social support strengthens the first stage of the indirect pathway from athlete engagement to social cohesion by improving interpersonal competence. Greater interpersonal competence, in turn, fosters stronger social cohesion through a stronger sense of belonging and trust.

### 4.4. Gender Differences in the Moderated Mediation Models

There were gender differences in the association between athlete engagement and team cohesion, but not in the mediating role of interpersonal competence. These results support **H4a** but do not support **H4b**.

Social support moderated the association between athlete engagement and team cohesion only in females. This pattern is consistent with social role theory, which holds that when gendered roles are affirmed by significant others, individuals form and enact different expectations and behaviors ([16]; [17]). For instance, females are often expected to display certain traits, such as cooperation, mutual care, and tolerance, whereas males may focus more on competition and individual performance when building team cohesion ([37]). These reasons may explain why social support did not moderate the direct prediction effect athlete engagement on team cohesion. In sport settings, male adolescents commonly exhibit stronger competitiveness and autonomous behavior and often face higher competitive pressure and performance expectations. In contrast, female adolescents in football seek to establish a sense of team belonging, experience acceptance within peer relationships, and receive positive affirmation from others. Therefore, males may view external support as a sign of doubt or challenge to their abilities rather than as a resource for team coordination. This perception can foster a tendency to avoid seeking support, which, in turn, may weaken the moderating role of social support in the association between athlete engagement and team cohesion. By contrast, female adolescents are more likely to interpret social support as a positive element of team interaction. This interpretation fosters trust, strengthens role identity, and aligns goals, which, in turn, may strengthen the association between athlete engagement and team cohesion. Furthermore, from the standpoint of biological evolution during human prehistory, females of most species, including humans, bore the primary responsibility for early parenting by conceiving, lactating, and caring for their offspring ([63]). As a result, female individuals were more comfortable being in a social group and forming strong team cohesion by befriending others to protect themselves and their offspring together ([63]).

Although previous studies have shown that social support differs in the quality of interpersonal relationship between male and female students ([12]; [35]), females tend to benefit from same-sex and teacher–student relationships to acquire more positive qualities, while males acquire more positive qualities from heterosexual relationships ([74]). However, this study did not distinguish the gender of the participants’ social contacts when investigating adolescent interpersonal competence. This may be part of the reason why there is no gender difference in the mediating effect of interpersonal competence.

### 4.5. Practical Implications

In practical terms, the conceptual model of group cohesion points to two key elements for school sport, namely clear goals and roles and a supportive climate. It is important to begin sessions with a short talk on goals and role division and to use shared performance cues during training and provide specific feedback after drills. When support from coaches and peers is credible and visible, athlete engagement is more likely to be expressed as coordinated action and collective follow through. These are core features of task cohesion. Schools sport should also build interpersonal competence and ensure that support channels are easy to use. For example, it is important to run brief communication drills and organize buddy pairs or peer mentoring. These activities strengthen communication, cooperation, and role identity. In a supportive environment, these behaviors foster trust and belonging and build shared commitment, which are the core elements of social cohesion.

Gender differences deserve attention. For male adolescents, it is important to place extra focus on team relationships and to make support visible with short daily conversations, peer mentoring or buddy pairs, and praise tied to specific behaviors. This approach helps interpret athlete engagement as a resource rather than a source of stress under pressure. For female adolescents, it is important to maintain a warm team climate and steady recognition. This can ease social evaluation worries and body image concerns, increase the team’s appeal, and support sustained engagement.

These recommendations can serve as a starting point for practice. Once effectiveness is verified in physical education, the approach can be expanded gradually to other subjects. For example, it is possible to use cooperative problem solving tasks so that students can clarify roles and plan strategies together. Teachers can run brief peer feedback and model support with encouragement plus specific suggestions. These steps help to build a positive team climate at the class and school levels, fostering belonging and emotional support, reducing anxiety and loneliness, and strengthening self-worth and identity. Strong social support networks improve coping, facilitate knowledge sharing and mutual learning, and can enhance academic performance, while shared norms promote positive behavior and reduce problem behaviors.

### 4.6. Limitations and Future Directions

This study has certain limitations. Firstly, the cross-sectional approach used in this study, while revealing correlations between the variables, does not allow for inference of causality between the variables tested ([49]). Future researchers could conduct longitudinal studies to determine whether the scores measured effectively represent an actual increase in team cohesion.

Secondly, this study did not distinguish the gender of the participants’ social contacts when investigating adolescent interpersonal competence. This may be one of the reasons why social support did not show gender differences in moderating the relationship between athlete engagement and interpersonal competence. Future research should consider differentiating the gender of participants’ social contacts to further clarify the moderated role of social support in the relationship between athlete engagement and interpersonal competence across different genders.

Thirdly, to comprehensively investigate the relationship between athlete engagement and team cohesion in adolescent groups, this study utilized a large sample covering all age groups of adolescents. However, demographic differences remain worthy of further exploration. Therefore, future research could focus on specific age ranges, such as primary school, middle school, or high school stages, to provide more precise evidence and support the development of personalized intervention strategies tailored to adolescents at different developmental stages.

## 5. Conclusions

This study enriches the conceptual model of group cohesion by testing a moderated mediation model and addressing gaps in existing research. The results showed that athlete engagement was positively associated with team cohesion. This relationship was moderated by social support, with gender differences observed. In addition, interpersonal competence partially mediated the link between athlete engagement and team cohesion. This indirect path was also strengthened by social support but showed no gender differences. These findings provide theoretical support for physical educators and highlight the value of strengthening team cohesion through team sports. Such efforts can promote adolescents’ social adaptation and mental health, helping them grow in a supportive and positive group environment.

## Figures and Tables

**Figure 1 behavsci-15-01264-f001:**
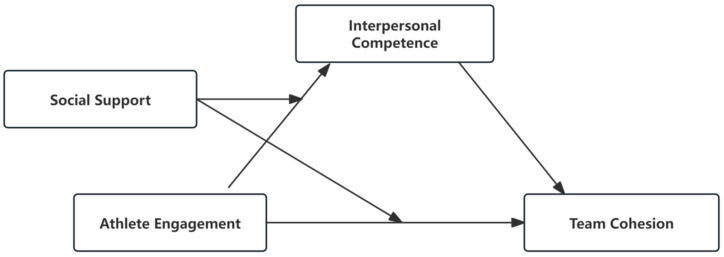
The moderated mediation model.

**Figure 2 behavsci-15-01264-f002:**
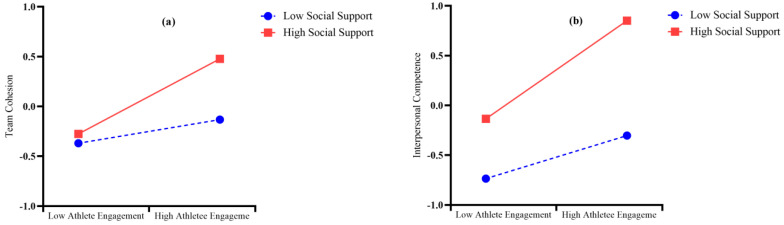
Moderating role of social support on the relationship between athlete engagement and team cohesion (**a**) and the relationship between athlete engagement and interpersonal competence (**b**).

**Figure 3 behavsci-15-01264-f003:**
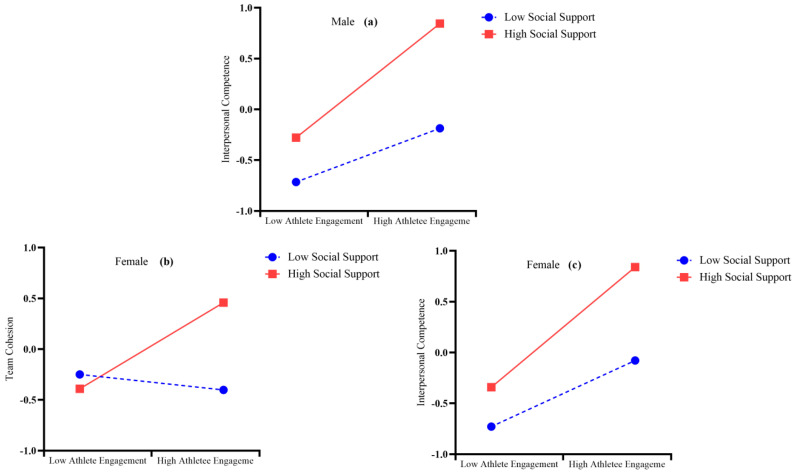
The moderating role of social support on the relationship between the relationship between athlete engagement and interpersonal competence for males (**a**). The moderating role of social support on the relationship between athlete engagement and team cohesion (**b**) and the relationship between athlete engagement and interpersonal competence for females (**c**).

**Table 1 behavsci-15-01264-t001:** Demographic information of participants.

Variables	Frequency	Percentage
**Gender**		
Male	1135	68.4%
Female	524	31.6%
**Level of school**		
Primary school students	881	53.1%
Junior high school students	543	32.6%
Senior high school students	231	14.3%
**Training per week**		
1~8 h	1047	63.1%
9~28 h	612	36.9%
**Football experience**		
1~5 years	1265	76.3%
6~12 years	394	23.7%

**Table 2 behavsci-15-01264-t002:** Correlation analysis among all variables.

Variables	*M*	*SD*	1	2	3	4
Athlete engagement	4.196	0.656	1			
2. Interpersonal competence	3.606	0.723	0.612 ***	1		
3. Social support	4.219	0.616	0.582 ***	0.575 ***	1	
4.Team cohesion	3.947	0.400	0.424 ***	0.425 ***	0.385 ***	1

Note: The mean (M) and standard deviation (SD) ***, *p* < 0.001.

**Table 3 behavsci-15-01264-t003:** Mediation of interpersonal competences (*n* = 1659).

	On Interpersonal Competence				On Team Cohesion			
	*β*	*SE*	*T*	95% CI	*β*	*SE*	*t*	95% CI
Athlete engagement	0.612	0.020	31.189 ***	[0.593, 0.672]	0.262	0.167	9.569 ***	[0.127, 0.193]
Interpersonal competence					0.265	0.167	9.666 ***	[0.125, 0188]
*R* ^2^	0.374				0.224			
*F*	991.524 ***				238.314 ***			

***, *p* < 0.001.

**Table 4 behavsci-15-01264-t004:** Moderated mediation model (*n* = 1659).

	On Interpersonal Competence				On Team Cohesion			
	*β*	*SE*	*T*	95% CI	*β*	*SE*	*t*	95% CI
Athlete engagement	0.438	0.228	19.890 ***	[0.409, 0498]	0.247	0.018	8.487 ***	[0.116, 0.186]
Interpersonal competence					0.172	0.017	5.897 ***	[0.068, 0.135]
Social support	0.354	0.017	16.031 ***	[0.249, 0.318]	0.175	0.013	6.218 ***	[0.057, 0.109]
Social support × athlete engagement	0.116	0.001	9.063 ***	[0.008, 0.012]	0.155	0.001	6.967 ***	[0.004, 0.008]
*R* ^2^	0.472				0.255			
*F*	494.460 ***				142.644 ***			

***, *p* < 0.001.

**Table 5 behavsci-15-01264-t005:** Moderated mediation model in the male group (*n* = 1135).

	On Interpersonal Competence				On Team Cohesion			
	*β*	*SE*	*t*	95% CI	*β*	*SE*	*t*	95% CI
Athlete engagement	0.398	0.030	15.372 ***	[0.405, 0524]	0.358	0.024	11.162 ***	[0.223, 0.318]
Interpersonal competence					0.177	0.022	5.443 ***	[0.076, 0.161]
Social support	0.337	0.022	13.708 ***	[0.259, 0.346]	0.149	0.017	4.822 ***	[0.050, 0.118]
Social support × athlete engagement	0.157	0.002	7.153 ***	[0.009, 0.016]	0.020	0.001	0.825	[−0.01, 0.004]
*R* ^2^	0.453				0.255			
*F*	313.806 ***				142.644 ***			

***, *p* < 0.001.

**Table 6 behavsci-15-01264-t006:** Moderated mediation model in the female group (*n* = 524).

	On Interpersonal Competence				On Team Cohesion			
	*β*	*SE*	*T*	95% CI	*β*	*SE*	*t*	95% CI
Athlete engagement	0.483	0.037	11.722 ***	[0.359, 0503]	0.221	0.172	3.440 ***	[0.025, 0.093]
Interpersonal competence					0.216	0.182	4.279 ***	[0.042, 0.114]
Social support	0.375	0.030	8.221 ***	[0.185,0.301]	0.272	0.130	3.700 ***	[0.023, 0.074]
Social support × athlete engagement	0.196	0.02	5.250 ***	[0.005, 0.122]	0.369	0.001	8.379 ***	[0.05, 0.007]
*R* ^2^	0.476				0.306			
*F*	157.202 ***				57.260 ***			

***, *p* < 0.001.

## Data Availability

The data that support the findings of this study are available from the corresponding author upon reasonable request.

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
