# Peer review of "Adolescent Athlete Engagement and Team Cohesion in Football: A Moderated Mediation Model with Gender-Based Insights"

_behavsci, 2025, doi:10.3390/bs15091264_

Round 1
Reviewer 1 Report
Comments and Suggestions for Authors
The following analysis is intended to support the reorientation of this valuable proposal toward dissemination spaces that uphold rigor and relevance:
Title
The title clearly reflects the central concepts: athlete engagement, team cohesion, moderated mediation, and gender differences. However, it contains redundancy and could benefit from greater conciseness. For instance, “in Football Sport” is unnecessarily repetitive—“football” already implies the sport. The phrase “Moderated Mediation Model and Gender Differences” is technically accurate but does not clearly convey the study’s original contribution.
Suggested revision:
Adolescent Athlete Engagement and Team Cohesion in Football: A Moderated Mediation Model with Gender-Based Insights
Abstract
The abstract is well structured and clearly presents the study’s objective, sample, instruments, results, and theoretical contributions. However, it lacks contextualization: why is this relationship relevant in adolescents and specifically in football? The sample is large (n = 1,692), but there is no mention of the sampling method or potential biases. The claim that the model “enriches” the Conceptual Model of Group Cohesion is vague and should be clarified. Additionally, the use of the Perceived Workplace Social Support Scale (PWSSS) raises methodological concerns regarding its suitability for adolescent athletes.
Suggestions:
- Add a brief justification for the study’s context.
- Reassess the appropriateness of the instruments for the age group and sport setting.
Keywords
The keywords are relevant but largely repeat terms from the title, which diminishes their indexing value. They also lack methodological and contextual terms that could enhance the article’s visibility.
Suggested additions:
- adolescents
- moderated mediation
- gender differences
- sports psychology
- group dynamics
Recommendation:
Consult academic thesauri such as the APA Thesaurus of Psychological Index Terms or the ERIC Thesaurus (ERIC) to select more standardized and specific terms.
General Recommendations
- Review grammatical and syntactic errors that affect clarity and professionalism (e.g., “stay and stay together”, “Interpersonal competence are the mediator”).
- Avoid vague or redundant phrases such as “success in all field” or “football involves a great deal of interaction”.
Introduction
The introduction follows a clear and progressive structure, beginning with the definition of group cohesion and leading to the formulation of hypotheses. The effort to ground the study in the Conceptual Model of Group Cohesion is appreciated, as is the inclusion of variables such as interpersonal competence, social support, and gender.
However, the section presents significant weaknesses in terms of scientific rigor, academic writing, and theoretical depth. It does not explain why football was chosen as the sport of focus, nor does it justify the relevance of studying adolescents in this context. The hypotheses are imprecisely formulated and contain grammatical errors (e.g., “Athlete engagement has a positive correlation with the team cohesion” should be “is positively associated with team cohesion”).
References
The references are relevant and up-to-date, though there is some inconsistency in theoretical depth. To strengthen the interpretation of youth engagement and cohesion in sport, it is recommended to include literature addressing the role of the family environment. In particular, the article “La adherencia del entorno familiar en el fútbol prebenjamín: un estudio de caso” (https://www.redalyc.org/pdf/3111/311148817014.pdf ) offers a valuable complementary perspective, highlighting how family involvement directly influences sports adherence and the formation of meaningful bonds in early childhood. A review of the family’s role is essential and can be justified through this reference.
Citation:
Merino, A., Arraiz, A., & Sabirón, F. (2017). La adherencia del entorno familiar en el fútbol prebenjamín: un estudio de caso. Revista Iberoamericana de Psicología del Ejercicio y el Deporte, 12(1), 139–148.
Method
The methodology section is clearly structured and includes essential elements: participants, procedure, instruments, and data analysis. The sample is large and the instruments used show high reliability. However, improvements are needed:
Suggestions:
- Expand on methodological limitations (e.g., convenience sampling, age range, lack of control for sociodemographic variables).
- Better justify the selection and adaptation of instruments.
- Explain how age heterogeneity was addressed.
- Clarify the role of each statistical technique in relation to the hypotheses.
Results
The results section is well organized and technically sound. The use of PROCESS models and simple slope tests is appropriate. The decision to analyze gender-based moderation effects adds depth to the model.
Points to consider:
- Demographic variables are not analyzed in relation to outcomes (e.g., urban vs. rural, educational level).
- Age range (8–16) is not segmented, limiting developmental interpretation.
- Common method bias is addressed, but no additional strategies are discussed.
- Minor language issues affect clarity (e.g., “the interaction was also a significant predictor…”).
- Gender differences are reported but not theoretically explored.
Discussion
The discussion follows the structure of the hypotheses and includes references to prior studies and cultural context, which is commendable. However, it is overly long and repetitive. The theoretical interpretation is limited, and there is no section on limitations or methodological constraints, which weakens the scientific value.
Recommendation:
Simplify the discussion, deepen theoretical reflection, and include a limitations section.
Conclusions
The conclusions are clearly structured and aligned with the study’s objectives. They effectively summarize the findings but could be more concise. A final reflection on practical implications, limitations, and future research directions would strengthen the section.
Comments on the Quality of English LanguageThe manuscript demonstrates an effort to communicate complex psychological and sociological constructs. The overall structure is coherent, and the technical terminology is generally used appropriately. However, the quality of English requires substantial revision to meet academic standards of clarity, precision, and professionalism.
In addition to general observations, several specific suggestions are provided throughout the detailed analysis of the manuscript, aimed at improving the clarity, accuracy, and academic tone of the text.
Reviewer 2 Report
Comments and Suggestions for Authors
This manuscript addresses an important question: how athlete engagement relates to team cohesion among adolescents in football, with attention to gender differences and a moderated mediation model. The topic is timely. In youth sport, questions of motivation, engagement, and group dynamics are central, and this paper has the potential to add valuable insights.
Clarity and structure
The manuscript is generally well-structured and easy to follow. The introduction sets the stage clearly, though in my view it could integrate more recent references to strengthen the rationale. The hypotheses are stated, but I think they could be positioned earlier and more explicitly to help the reader. The writing is clear, although some sentences are overly long and could be simplified.
Methodology
The study sample is adequate, and the focus on adolescents is well justified. However, I would like to see more detail on the recruitment process. For example, how were the teams selected, and could there be any selection bias? The measures used (engagement, cohesion, etc.) are appropriate and widely recognized, but the description of their validation in the specific cultural context is quite brief. In my opinion, expanding this would reassure readers that the tools truly captured the constructs among this population.
Statistical analysis
The moderated mediation model is suitable for the research questions, and the analysis seems competently conducted. Still, the manuscript does not discuss checks of model assumptions or possible multicollinearity. It would be important to mention whether the results are robust under different specifications. Also, I recommend that the authors explain more clearly how they tested for gender differences currently, this section reads as a bit too technical without sufficient interpretation.
Results and interpretation
The findings are interesting, especially the gender-related differences. The interpretation is reasonable, but at times it feels too definitive. For example, some claims about causal mechanisms go beyond what the cross-sectional design can support. I think the authors should acknowledge more explicitly the limits of inferring causality. Also, the discussion could better connect the findings to broader theories of youth motivation and group dynamics.
Strengths
- Relevant and timely topic.
- Clear focus on adolescents, an underexplored group in this context.
- Use of a sophisticated statistical model that allows for nuanced insights.
Weaknesses and limitations
- Limited detail on sampling and potential biases.
- Sparse discussion of the cultural validity of measurement tools.
- Some results interpreted too strongly given the design.
- Missing discussion of model diagnostics and robustness.
Suggestions
I recommend the authors:
- Expand the methodological section with more detail on sample recruitment and validation of measures.
- Report (or at least briefly mention) statistical assumption checks and robustness tests.
- Soften causal language and emphasize the cross-sectional limits.
- Strengthen the discussion by linking results to existing motivational and team dynamics theories.
- Consider including more recent references to position the work within current debates.
Conclusion
In my opinion, this is a solid and promising manuscript. The topic is meaningful, the design is sound, and the findings are potentially useful for both researchers and practitioners. With some refinements in the methods description and a more cautious interpretation, I believe the paper could make a valuable contribution.
Reviewer 3 Report
Comments and Suggestions for Authors
This manuscript addresses a timely and relevant topic within youth sport psychology, offering a thoughtful investigation into the relationship between group cohesion and psychological need satisfaction, with gender as a moderating factor. The study is methodologically sound and presents clear findings that could be of interest to both researchers and practitioners. To further refine the manuscript and enhance its scholarly impact, I suggest the following revisions:
< !--StartFragment -->
1. Clarify the originality and theoretical contribution: while the use of a moderated mediation model is innovative, this aspect could be more explicitly framed in the introduction.
< !--StartFragment -->2. Improve structure and clarity in key sections: some paragraphs, particularly in the introduction and conclusion, are dense and could benefit from clearer transitions and more concise phrasing.
< !--StartFragment -->
3. Engage more deeply with recent international literature: the manuscript references foundational studies, but it would benefit from integrating more recent international research (post-2020) to strengthen its relevance.
< !--StartFragment -->
4. Expand the discussion of practical implications: the discussion section could better articulate how the findings translate into actionable insights for coaches, educators, etc.
< !--EndFragment -->
< !--StartFragment -->
5. Ensure consistency and accuracy in references: the references are generally appropriate, but some appear inconsistent in formatting or slightly outdated:
check APA formatting (e.g., italics for journal titles) and verify that all cited studies are included in the reference list. Also, consider updating outdated citations.
< !--EndFragment -->
< !--EndFragment -->
< !--EndFragment -->
Reviewer 4 Report
Comments and Suggestions for Authors
Thank you for your submission to the journal. I hope that the reviews below will be revised to the level of publication. Good luck to all the authors.
1. Introduction
Required revision: A complete restructuring of the introduction is required
1) Not following the development flow of a general paper introduction (research background → research necessity → explanation of research-related variables → pattern of previous research → differentiation from previous research → need to be structured as the purpose and expected effect of the study)
2) There is no citation of any previous research that deals with Athlete Engagement as an independent variable, and there is a lack of theoretical justification for setting Athlete Engagement as an independent variable in this study
3) The originality of the research is not sufficiently emphasized, Lack of presentation of differentiation of the subject compared to previous research
2. Theoretical Background
Required revision: A complete reconstruction of the theoretical background is required
1) There are no citations of previous studies that have dealt with athlete engagement as an independent variable.
2) Lack of existing research and theoretical justification to support the model of 'exercise engagement→ interpersonal competence, → team cohesion'
3) Insufficient evidence for setting social support as a moderating variable (weak theoretical explanation, level of simple listing of previous studies)
4) It is necessary to present sufficient content for the previous research for each set variable (independent variable, parameter, dependent variable)
3. Research Methods
Required revision: Present all survey questions used
1) It is difficult to determine the causal relationship with the actual content because the questionnaire questions and scales used are not specifically presented. → It is necessary to present what kind of content is present in the questionnaire questions.
4. Results
There are no special problems
5. Discussion
Required revision: Practical utilization plan and in-depth discussion need to be supplemented
1) There are some comparisons of results with previous studies, but the interpretation of the results is fragmented and few practical applications (coaching, education, policy application, etc.) are proposed
2) Lack of in-depth discussion on the main factors of the research model, such as social support and gender differences
6. Conclusion
Required revision: Need to construct and include empirical implications
1) Only focus on summarizing research results, No practical implications at all→ Lack of presentation of field applicability, failing to convince readers of the importance and necessity of research.
Round 2
Reviewer 4 Report
Comments and Suggestions for Authors
Dear authors,
You have worked hard to carry out the revision. However, reinforcement of the introduction is still needed. The content that needs to be revised is mentioned below, so please proceed with the revision to increase the academic value.
- Adequate description of the social environment
In the current introduction, only the social attributes of team sports are briefly mentioned. It is necessary to explain what kind of social environment is related to the topic of this study and why the content of this study is important. Furthermore, it is necessary to explain the relationship between adolescent athletes and team sports.
- Abstract explanation of the relationships among variables
The study does not provide a concrete explanation of why the variables (Athlete Engagement, Interpersonal Competence, and Team Cohesion) influence one another or through which mechanisms these relationships operate. In addition, a more detailed explanation is needed regarding why Social Support serves as a moderating and mediating factor and what practical significance this has in the context of real team sports settings.
- Lack of specificity in identifying limitations of previous studies
The current introduction only states in general terms that prior research has lacked sufficient discussion. It is necessary to describe what points were lacking through citations of specific examples of previous studies.
This kind of elaboration can help to emphasize the significance of the present study.
